

# Ecological divergence of a mesocosm in an eastern boundary upwelling system assessed with multi-marker environmental DNA metabarcoding

Markus A. Min[1,2], David M. Needham[1,3], Sebastian Sudek[1], N. Kobun Truelove[1], Kathleen J. Pitz[1],
Gabriela M. Chavez[1,3], Camille Poirier[1,3], Bente Gardeler[1,3], Elizabeth von der Esch[4], Andrea
Ludwig[3], Ulf Riebesell[3], Alexandra Z. Worden[1,3], and Francisco P. Chavez[1]

[1]Monterey Bay Aquarium Research Institute, Moss Landing, United States of America
[2]University of Washington School of Aquatic and Fishery Sciences, Seattle, United States of America
[3]GEOMAR Helmholtz Centre for Ocean Research Kiel, Kiel, Germany
[4]Institute of Hydrochemistry, Technical University of Munich, Germany

*Correspondence to*: Francisco Chavez (chfr@mbari.org)



**Abstract**

Eastern boundary upwelling systems (EBUS) contribute a disproportionate fraction of the global fish catch relative to their size and are especially susceptible to global environmental change. Here we present the evolution of communities over 50 days in an *in situ* mesocosm 6 km offshore of Callao, Peru and in the nearby unenclosed coastal Pacific Ocean.

The communities were monitored using multi-marker environmental DNA (eDNA) metabarcoding and flow cytometry. DNA extracted from weekly water samples were subjected to amplicon sequencing for four genetic loci: 1) the V1-V2 region of the 16S rRNA gene, for photosynthetic eukaryotes (via their chloroplasts) and bacteria; 2) the V9 region of the 18S rRNA gene for exploration of eukaryotes but targeting phytoplankton; 3) cytochrome oxidase I (COI), for exploration of eukaryotic taxa but targeting invertebrates, and 4) the 12S rRNA gene, targeting vertebrates.

The multi-marker approach showed a divergence of communities (from microbes to fish) between the mesocosm and the unenclosed ocean. Together with the environmental information, the genetic data furthered our mechanistic understanding of the processes that are shaping EBUS communities in a changing ocean. The unenclosed ocean experienced significant variability over the course of the 50-day experiment with temporal shifts in community composition but remained dominated by organisms that are characteristic of high nutrient, upwelling conditions (e.g.

diatoms, copepods, anchovies). A large directional change was found in the mesocosm community. The mesocosm community that developed was characteristic of upwelling regions when upwelling relaxes and waters stratify (e.g. dinoflagellates, nanoflagellates). The selection of dinoflagellates under the warm (coastal El Niño) and stratified conditions in the mesocosm may be an indication of how EBUS will respond under the global environmental changes (i.e. continued global warming) forecast by the IPCC.

**1 Introduction**

Eastern boundary upwelling systems (EBUS) are exceptionally productive marine ecosystems: they account for 5% of total global primary production (Carr, 2001) and 20% of marine fish production (Chavez and Messié, 2009) while occupying less than 1% of the area of the ocean. Strong physical forcing drives productivity in these ecosystems; upwelling-favourable winds bring macro- and micronutrients from depth to the surface. Under favourable temperature

and light conditions phytoplankton bloom (Messié and Chavez, 2015), leading to increases in biomass in higher trophic levels (Chavez and Messié, 2009; Ayón et al., 2008). However, these systems are variable both physically and ecologically, making it difficult to develop mechanistic understanding of the links between ecological, biogeochemical, and physical processes. Given that these systems may be disproportionately affected by climate change (Gruber, 2011) it is key that we develop predictive understanding of how these systems will change over time.

Here we present results from a perturbation experiment geared at targeting this problem.

Observational studies provide insights into processes regulating biological production and community structure in upwelling systems, but because of the complex interplay of multiple factors, it is difficult to assess the relative contributions of drivers causing change. The EBUS literature has extensive analyses of correlative relationships in the

pursuit of causative understanding (e.g. Carr and Kearns, 2003; Messié and Chavez, 2015; Patti et al., 2008). By



isolating natural communities in enclosures or mesocosms, one can physically perturb the system in a controlled manner (Stewart et al., 2013; Riebesell et al., 2008; Riemann et al., 2000; Sandaa et al., 2009), providing a method for studying mechanisms driving responses of these systems to perturbations. However, contained mesocosms remove horizontal mixing processes, can modify vertical mixing, and remove top predators (fish and mammals), and thus are

not exact analogues of natural marine ecosystems.

Variations in zooplankton, phytoplankton, and/or bacteria have been monitored in mesocosm experiments using a variety of sampling techniques, including nets of various mesh size, direct counts of bacteria from water samples, flow cytometry for enumeration of small phytoplankton and bacteria, or algal pigment analysis (e.g. Hitchcock et al., 2016;

Suffrian et al., 2008). Environmental DNA (eDNA) metabarcoding is a complementary, rapidly evolving biomonitoring technique that can survey these communities by examining both extracellular and intracellular DNA present in environmental samples (Taberlet et al., 2012). Here we define eDNA as any DNA captured by filtering seawater through a low porosity filter (Chavez et al. 2021) and as such it includes intact microbial cells and other small live organisms as well as material shed or produced by larger plants and animals that has not yet degraded. By

targeting and amplifying a highly variable region of the genome across numerous taxonomic groups, eDNA metabarcoding allows for the simultaneous detection and identification of a diversity of taxa (Valentini et al., 2016) and has been used in a variety of aquatic settings and across a wide range of organisms, recovering greater alpha diversity than visual counts or morphological identification (Djurhuus et al., 2018; Boussarie et al., 2018). While eDNA metabarcoding has been used in mesocosms to demonstrate its effectiveness by detecting and identifying

known species assemblages (Kelly et al., 2014; Evans et al., 2016), its utility to detect change across multiple trophic levels has not been demonstrated in perturbation mesocosm experiments, nor have multiple eDNA markers been used simultaneously to monitor community dynamics. By providing information about broad taxonomic groups, eDNA metabarcoding provides a holistic, community-level view of ecological changes occurring within mesocosms.

Here, we present results from an *in situ* mesocosm experiment that took place in austral summer 2017 in the coastal Peruvian upwelling system near Callao (Bach et al., this issue). To study how marine populations and biogeochemical properties change during an upwelling event, nutrient-rich water collected in the regional oxygen minimum zone (OMZ) was added to the mesocosms. The mesocosms were later modified by the injection of a salt brine solution to maintain the vertical density gradient and prevent full water column mixing. This resulted in heavily stratified

mesocosms. A third, unintended perturbation occurred when seabirds began to hover over and perch on the mesocosms during the last month, further modifying biogeochemical and ecological conditions. Alongside the core physicochemical measurements to characterize environmental conditions, flow cytometry and eDNA samples were taken both in the surface waters of the nearby unenclosed Pacific Ocean at the highly variable and dynamic study site and from a mesocosm over the 50-day experiment period. Four genetic loci – the V1-V2 region of the 16S rRNA gene

(Sudek et al., 2015; Giovannoni et al., 1990), the V9 region of the 18S rRNA gene (Amaral-Zettler et al., 2009; Stoeck et al., 2010; Amaral-Zettler et al., 2018), mitochondrial cytochrome oxidase I (COI) (Leray et al., 2013; Folmer et al.,



1994), and the mitochondrial 12S rRNA gene (Miya et al., 2015) were evaluated, capturing a diversity of bacterial, phytoplankton, zooplankton, and vertebrate populations, respectively.

Using this multi-marker approach, we detected the ecological divergence of the mesocosm relative to the nearby unenclosed ocean. The mesocosm communities evolved to be dominated by taxa typical of stratified conditions, whereas the unenclosed ocean retained a community shaped by high nutrients and intermittent upwelling. The impact of resting seabirds was detected via a "orni-eutrophication" driven phytoplankton bloom (Bach et al., this issue) and the appearance of DNA of fish, seabirds, and bacteria typical of host-association of animal microbiomes

(Saccharibacteria) (Jaffe et al., 2021). This study reveals a clear community shift driven by an experimental manipulation that simulated upwelling and subsequent stratification, as well as the impact of external inputs to the experimental system, and demonstrates that eDNA metabarcoding is a powerful tool for detecting community-level changes over time.

## 2 Materials and methods

### 110 2.1 Mesocosm deployment and manipulations

On February 22, 2017, eight "Kiel Off-Shore Mesocosms for Ocean Simulations" (Riebesell et al., 2013) were deployed just north of Isla San Lorenzo, 6 km off the Peruvian coastline (Fig. 1). Each mesocosm consisted of a cylindrical 18.7 m long polyurethane bag (2 m diameter, 54.4 ±1.3 m$^3$ volume) suspended in an 8 m tall flotation frame (Bach et al., this issue). After allowing water exchange for three days through nets (mesh size 3 mm) at the top

and bottom of each mesocosm, the water mass inside each of the mesocosms was isolated from the surrounding water by attaching a conical sediment trap to the lower end of the bags and pulling the upper ends of the bags ~1.5 m above the surface. The enclosing of the mesocosms marked the start (day 0) of the 50-day experiment. Sampling of chl-a and physicochemical and biogeochemical conditions was performed on all 8 mesocosms as well as in the nearby coastal ocean within a few meters from the mesocosms. The eDNA and flow cytometry samples were collected from

all mesocosms but only analysed for mesocosm M1 and the nearby unenclosed coastal Pacific Ocean, where samples were taken within a few meters of the mesocosm frame. These two collections and their analysis are referred to as mesocosm and Pacific hereafter.


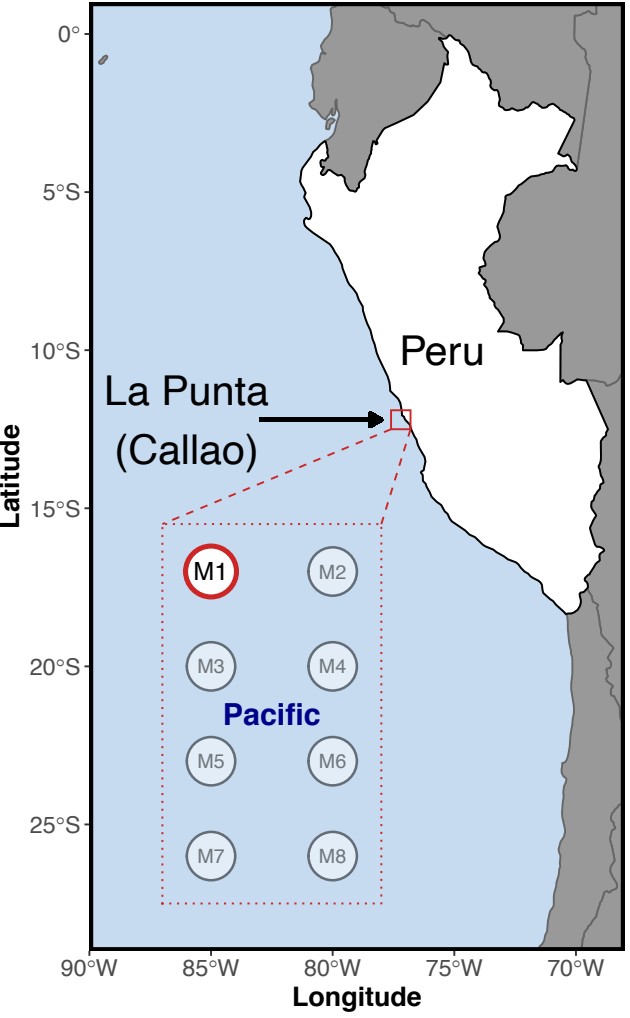

**Figure 1. The KOSMOS 2017 study site. The mesocosms were deployed 6 km offshore of La Punta (Callao), just north of Isla San Lorenzo (12.0555°S; 77.2348°W). All mesocosms were sampled for chl-a, various biogeochemical variables, and multiple physicochemical parameters (Bach et al., this issue). Mesocosm M1 (highlighted in red) was also sampled for DNA and flow cytometry, as was the nearby unenclosed Pacific Ocean. Please note that the square marking the study site is not true to scale.**

Over the course of the experiment, three primary intentional manipulations took place in all 8 mesocosms: OMZ water addition, salt additions to control stratification, and additions of organisms. On days 11 and 12 of the experiment, a total of ~20 $m^3$ of OMZ water collected at a depth of 30 m was exchanged with the water enclosed in M1; this water had been collected on day 5 from the OMZ located at station 1 (12.028323°S; 77.223603°W) of the IMARPE (Instituto del Mar del Perú) time-series transect (Graco et al., 2017) using deep water collectors (Taucher et al., 2017). In order to keep the mesocosm stratified and thus preserve the low $O_2$ bottom layer, a NaCl brine solution was injected evenly into the bottom layers of the mesocosm on days 13 (69 L at a depth range of 10 – 17 m) and 33 (46 L at a depth range



of 12.5 – 17 m) of the experiment. Two endemic organisms, larvae of the Peruvian Scallop (*Argopecten purpuratus*) and eggs of the Fine Flounder (*Paralichthys adspersus*) were added to all mesocosms. On day 14, Peruvian Scallop larvae was added in concentrations of ~10,000 individuals m$^{-3}$, and on day 31 Fine Flounder eggs were added in concentrations of ~90 individuals m$^{-3}$. However, few scallop larvae and no fish larvae were detected during subsequent

sampling via vertical tows of an Apstein net of mesh size 100 μm (unpublished data), and Fine Flounder DNA was only detected in the two samples directly following the addition, indicating that they either degraded or sank out of the surface layer.

## 2.2 Sample collection

Salinity, temperature, and chl-a fluorescence were measured with vertical casts of a CTD60M sensor system (Sea &

Sun Technologies) in all mesocosms and the Pacific. Samples for inorganic nutrients were collected using a 5 L "integrating water sampler" (IWS) (Hydro-Bios Kiel) that evenly collected water for two separate depth ranges, the surface and bottom waters. However, the inorganic nutrient, chl-a, temperature, and salinity results presented in the first column of Figure 2 were calculated by averaging the IWS-collected data over the two depth ranges. After transport back to an on-shore laboratory, nutrient samples were filtered (0.45 μm filter, Sterivex, Merck) and analysed using an

autosampler (XY2 autosampler, SEAL Analytical) and a continuous flow analyser (QuAAtro AutoAnalyzer, SEAL Analytical) connected to a fluorescence detector (FP-2020, JASCO). Silicic acid (Si(OH)$_4$) was analysed colorimetrically following the procedures by Mullin and Riley, (1955). Nitrate (NO$_3^-$) and nitrite (NO$_2^-$) were quantified through the formation of a pink azo dye as established by Morris and Riley (1963). Note that other measurements were made throughout the experiment from all mesocosms as reported elsewhere (Bach et al., this

issue). CTD casts and IWS water collections occurred every other day (except for days 1-4 and 12-18, when they were taken daily),

DNA samples were taken in M1 and the Pacific on days 1, 8, 15, 24, 32, 36, 42, and 48. Flow cytometry samples were collected roughly every other day. Water for eDNA and flow cytometry measurements was collected using the IWS

to evenly sample the upper portion of the water column; the depths sampled were 0 – 5 m on days 1 and 2, 0 – 10 m from day 3 to 28, and 0 – 12.5 m from day 29 to 50. After transport back to shore, eDNA samples were collected by filtering 250 mL of water onto a 47 mm diameter 0.22 μm pore size polyvinylidene difluoride membrane filter (Millipore, USA) using a vacuum pump. All filters were flash frozen in liquid nitrogen and stored at -80°C before being shipped on dry ice to California, USA for analysis.


Field controls consisting of 250 mL of filtered reverse osmosis (RO) and MilliQ water were collected on Day 3 of the experiment in order to characterize any contaminating taxa in these systems and later steps. No DNA was detected in these controls using NanoDrop 1000 spectrophotometer (ThermoFisher Scientific, Waltham, MA) measurements; environmental samples ranged from 19.9 – 139.3 ng/μL. However, COI, 18S, and 16S rRNA PCR amplification

yielded slight amplification and PCR products were sequenced (Table S1). After normalization steps, the dominant reads in the bacterial portion of the 16S rRNA sequences consisted of Betaproteobacteria (36.2%), that were not





prominent in field samples (0.27% mean proportion of reads in field samples). The greatest proportion of plastidial reads in the field blanks were multiple diatom ASVs (66.7% of reads), with the same ASVs composing 42.8% of reads in the field samples. In the COI reads, most ASVs found in the field blanks were also not prominent in the field

samples, with the exception of the calanoid copepod *Paracalanus* (12.4% of reads in field blanks, 13.3% of reads in field samples). In the 18S reads, the top ASVs in the field blanks were the copepods *Hemicyclops thalassius* (30.0% of reads in field blanks, 3.9% of reads in environmental samples) and *Paracalanus* (11.7% of reads in field blanks, 14.6% of reads in environmental samples). While this indicates that there may have been a low level of field-based cross-contamination in our environmental samples (a near inevitability when sampling in a remote location without a

dedicated molecular laboratory), these reads were not removed from our analyses.

### 2.3 Flow cytometry

Triplicate samples of 1 mL volume each were taken and preserved with glutaraldehyde (EM grade, final concentration of 0.25%). The samples were then incubated for 20 minutes in the dark and subsequently treated the same way as the

eDNA samples. Photoautotrophs (photosynthetic eukaryotes, *Synechococcus*, and *Prochlorococcus* were it present) and heterotrophic bacteria were enumerated on a BD Influx cell sorter (BD Biosciences, USA) equipped with a 488-nm argon laser (200 mW) and a 70 μm nozzle running with 0.2 μm pre-filtered 1xPBS (10xPBS, pH 7.4, Life Technologies). Prior to the running of each sample, fluorescent polystyrene beads (0.75 μm yellow green beads, Polysciences, Inc) were added for reference. For calculation of the total volume analysed, the samples were weighed

before and after each run. For counts of photoautotrophs, the system was triggered on forward angle light scatter (FALS), and red chlorophyll autofluorescence (692/40 nm band-pass filter) as well as orange phycoerythrin autofluorescence (572/27 nm band-pass filter) versus FALS and were recorded over 8 min. running at ~25 μL min$^{-1}$. To count heterotrophic bacteria, the samples were stained with SYBR Green I (10,000x SYBR Green I, Thermo Fisher; final concentration of 0.5x with 15-minute incubation time in the dark) and was triggered on green fluorescence

(520/35 nm band-pass filter). The samples were run for 1 min. at ~25 μL min$^{-1}$. FCS-files were processed in WinList 3D 9.0.1 (Verity Software House, Topsham ME, USA). Among other bacterial populations, a unique population of presumed bacteria appeared in mesocosm and coastal samples after day 24 of the study and was gated in accordance with the representative cytogram (Fig. S1).

### 2.4 DNA extraction

DNA from filters was extracted using the DNeasy® Blood and Tissue kit (Qiagen, Germantown, MD) following standard protocol with some modifications that included an overnight incubation and increasing the amount of lysis buffer to completely submerge the filter during lysis (protocols.io: dx.doi.org/10.17504/protocols.io.xjufknw). DNA extraction concentrations were quantified using NanoDrop 1000 spectrophotometer (ThermoFisher Scientific, Waltham, MA) measurements.


### 2.5 DNA amplification and sequencing



### 2.5.1 Cytochrome oxidase I (COI), 18S rRNA, and 12S rRNA

PCR reactions for COI and the 12S rRNA gene were run with Fluidigm two-step amplification protocol for each sample (COI protocols.io: dx.doi.org/10.17504/protocols.io.mwnc7de, 12S protocols.io: dx.doi.org/10.17504/protocols.io.bcppivmn), while PCR reactions for the 18S rRNA gene were run using 12-basepair Golay barcoded reverse primers (protocols.io: dx.doi.org/10.17504/protocols.io.n2vdge6). For COI, the primary PCR primers used are from Leray et al., 2013 and are as follows: Fluidigm CS1+**mlCOIinfF** (forward): ACACTGACGACATGGTTCTACA **GGWACWGGWTGAACWGTWTAYCCYCC** and Fluidigm CS2+**HCO2198** (reverse): TACGGTAGCAGAGACTTGGTCT **TAAACTTCAGGGTGACCAAAAAATCA.** For 18S, primary PCR primers used are from Amaral-Zettler et al., 2009 and are as follows: Euk1391F (forward): AATGATACGGCGACCACCGAGATCTACAC TATCGCCGTT CG **GTACACACCGCCCGTC** and EukBr (reverse): CAAGCAGAAGACGGCATACGAGAT XXXXXXXXXXXX AGTCAGTCAG CA **TGATCCTTCTGCAGGTTCACCTAC** (where XXXXXXXXXXXX is unique 12-bp barcode location, all primers listed in 5' to 3' direction). For 12S, primary PCR primers used are from Miya et al., 2015 and are as follows: Fluidigm CS1 + **12S MiFish_U** (forward): ACACTGACGACATGGTTCTACA**GTCGGTAAAACTCGTGCCAGC** and Fluidigm CS2 + 1**2S MiFish_U** (reverse): TACGGTAGCAGAGACTTGGTCT**CATAGTGGGGTATCTAATCCCAGTTTG**.

For each of the three markers, primary PCR amplifications were carried out in triplicate 25 µL reactions using 1 µL DNA extract, 12.5 µL AmpliTaq Gold Fast PCR master mix (Applied Biosystems), 1 µL each of forward and reverse primers (5 µM), and 9.5 µL molecular-biology grade water. PCR reactions were run in 96-well plates with a NTC run in triplicate for each plate. Primary COI cycling parameters were 95 °C for 10 min followed by 16 cycles of 94 °C for 10 seconds, 62 °C for 30 seconds, 68 °C for 60 seconds, next followed by 25 cycles of 94 °C for 10 seconds, 46 °C for 30 seconds, 68 °C for 60 seconds, and a final step of 72 °C for 10 min. Primary 18S rRNA cycling parameters were 95°C for 10 min followed by 35 cycles of 94 °C for 45 seconds, 57 °C for 30 seconds, 68 °C for 90 seconds, and a final elongation step of 72 °C for 10 min. Primary 12S rRNA cycling parameters were 95 °C for 15 min followed by 13 cycles of 94 °C for 30 seconds, 69.5 °C for 30 seconds (changes -1.5°C per cycle), 72 °C for 90 seconds, next followed by 25 cycles of 94 °C for 30 seconds, 50 °C for 30 seconds, 72 °C for 45 seconds, and a final step of 72 °C for 10 min.

Following PCR, the pooled PCR products for each genetic marker were run through an agarose gel to confirm the presence of target bands and inspected for degree of amplification, as well as absence of any non-specific amplification. PCR products were purified and size selected using the Agencourt AMPure XP bead system (Beckman Coulter, USA). A second agarose gel was run to confirm primer removal and retention of target amplicons after purification.

Library preparation and sequencing was conducted at the Research Technology Support Facility (RTSF) Genomics Core at Michigan State University (MSU) as was secondary amplification for COI and 12S. PCR products were run



through Invitrogen SequalPrep Normalization Plate (ThermoFisher Scientific) using the manufacturer's protocol to
create pooled libraries. The pooled product was loaded on a standard MiSeq v2 flow cell and sequenced in a 2x250bp
(COI, 12S rRNA) or 2x150 bp (18S rRNA) paired end format using a v2 500-cycle MiSeq reagent cartridge. The
MiSeq run was performed with a 10% PhiX spike. Custom sequencing primers were added to appropriate wells of the
reagent cartridge. Base calling was done by Illumina Real Time Analysis (RTA) v1.18.54 and output of RTA was
demultiplexed and converted to FastQ format with Illumina Bcl2fastq v2.18.0.

**2.5.2 16S rRNA**

Prior to amplification, DNA was diluted to 5 ng $\mu L^{-1}$ with TE pH 8. The V1-V2 16S rRNA gene region was amplified
as previously described (Sudek et al., 2015) with 5 $\mu L$ of 10× buffer, 1 U of HiFi-Taq, 1.6 mM MgSO4, 5 ng of
template DNA, and 200 nM of 27F (AGRGTTYGATYMTGGCTCAG, Daims et al., 1999) and 338RPL primer
(GCWGCCWCCCGTAGGWGT, Morris et al., 2002). PCR cycling parameters were 95 °C for 2 min, 30 cycles of 94
°C for 15 seconds, 55 °C for 30 seconds and 68 °C for 1 min, followed by a final elongation at 68 °C for 7 min.
Purification, barcoding, library preparation, and sequencing were performed at University of Arizona with MiSeq
2x300 bp reads.

**2.6 Bioinformatics**

**2.6.1 Cytochrome oxidase I (COI), 18S rRNA, and 12S rRNA**

The resulting Illumina sequence data was analysed through a custom shell script adapted from the banzai pipeline
(https://github.com/MBARI-BOG/BOG-Banzai-Dada2-Pipeline, O'donnell et al., 2016). Complete script and
parameters are included in the Supplement. Within the script, primer sequences were first removed from fastq files
through the program Atropos (Didion et al., 2017). Fastq files were then fed into the DADA2 program (Callahan et
al., 2016). DADA2 models error on a per-Illumina run basis, controlling for read quality and picking amplicon
sequence variant (ASV) sequences that represent biological variability rather than sequencing error (Callahan et al.,
2016). Within DADA2, reads were trimmed to remove low-quality regions and filtered by quality score, sequencing
errors were modelled and removed, and reads were then merged and chimeric sequences removed. Taxonomy was
assigned to the resulting ASV sequences through blastn searches to NCBI GenBank's non-redundant nucleotide
database (nt) (Camacho et al., 2009; Agarwala et al., 2018). Blast results were filtered using MEGAN6's lowest
common ancestor (LCA) algorithm (Huson et al., 2016). Only hits with ≥80% sequence identity, ≥100 bitscore and
whose bitscores were within the top 2% of the highest bitscore value for each ASV were considered by MEGAN6.
The MEGAN6 parameter LCA percent was from 0.80 to 0.85, depending on the marker, allowing for up to 15 – 20%
of top hits to be off target and still have the majority taxonomy assigned. This parameter value was chosen to allow
for minor numbers of incorrectly annotated GenBank entries – effectively allowing for ASVs which had many high-
quality hits to a taxa to still be assigned to that taxa even if there existed a high-bitscore hit to another GenBank
sequence annotated to an unrelated taxa. We decided this was more advantageous than the disadvantage caused by
ignoring small numbers of true closely related sequences. Furthermore, post-MEGAN6 filtering was performed to
ensure only contigs with a hit of ≥97% sequence identity and ≥200 bitscore were annotated to the species level. Only





contigs with a hit of ≥95% sequence identity and ≥150 bitscore were annotated to the genus level. Annotations were
elevated to the next highest taxonomic level for contigs that failed those conditions.

### 2.6.2 16S rRNA

Demultiplexed reads were imported into QIIME2 (Bolyen et al., 2019) and cutadapt (Martin, 2011) trim-paired was
used to trim primers. Trimmed reads were denoised with DADA2 (Callahan et al., 2016) denoise-paired command,
with a truncation of the forward and reverse reads to 250 and 225 bp, respectively. Resulting ASVs were classified in
QIIME2 with the feature-classifier classify-consensus-blast (Camacho et al., 2009) command with a percent identity
of 0.95, maximum number of accepted hits of 5 and consensus of 0.7 against the 99% representative sequences of
SILVA 132 (Quast et al., 2013) and both 18S and 16S rRNA gene references. A bacterial ASV table was then
generated by removing mitochondrial and plastidial, sequences. Plastidial sequences were then selected based on the
SILVA classification and reclassified with the Phytoref database (Decelle et al., 2015) using the same settings as
before except with a percent-identity of 0.90. The five most abundant cyanobacterial ASVs were further analysed by
manual blast against NCBI-nt excluding environmental sequences.

### 2.7 Quality Control and Decontamination

Following the bioinformatic pipeline, the COI, 18S rRNA, and 12S rRNA sequencing results were passed through
custom R ver. 3.6.0 (R Core Team, 2019) decontamination scripts. For each plate, we first removed all singleton
ASVs. Next, for each ASV that was detected in at least one of the PCR blanks on the plate, we determined the
maximum number of reads of that ASV in any of the individual PCR blanks and subtracted this value from the reads
of the ASV in each of the environmental samples. This was done to address cross-contamination from various sources,
such as tag-jumping (Schnell et al., 2015). As mentioned previously, no decontamination steps were taken using the
field blank samples. Finally, we removed all reads assigned to common terrestrial contaminants: orders Rodentia and
Lagomorpha, families Hominidae, Bovidae, Felidae, and Canidae, and genera *Gallus* and *Meleagris*. These
decontamination steps were not run for 16S rRNA sequences because the PCR blanks were not sequenced.

### 2.8 Statistical analyses

Beta diversity analyses were run separately on the five datasets (COI, 18S rRNA, 12S rRNA, 16S rRNA bacterial
sequences, and 16S rRNA plastidial sequences) using the QIIME2 (Bolyen et al., 2019) DEICODE plugin. The beta
diversity analyses for 12S rRNA are included in the Supplement (Fig. S2) since vertebrates were excluded from the
mesocosm (water filtered through a 3 mm mesh), and vertebrate eDNA that was detected was: 1) found during our first
sampling and then decayed and disappeared at a rate consistent with experimental results (Sassoubre et al., 2016); 2)
result of the intentional addition of fish eggs, and 3) the unintentional addition from seabird faeces. Through matrix
completion and robust Aitchison PCA (RPCA), DEICODE (Martino et al., 2019) is particularly well-suited to handle
the sparseness inherent to sequencing data. A PERMANOVA test was also run in QIIME2 on the Aitchison distance
matrix produced by DEICODE to determine the significance of the variance between M1 samples following OMZ
water addition and the other samples (Pacific and first two M1 samples). Based on the results of the RPCA and



PERMANOVA, which showed a clear differentiation between the M1 samples following OMZ water addition (day 15 and on) and Pacific surface water samples on PC1 for the four primary datasets analyzed for beta diversity (all but

12S rRNA), the loading scores on PC1 were used to identify ASVs that were driving the divergence between M1 and the Pacific. The PC results presented are constrained to ASVs that were consistently present throughout the experiment as determined by ASVs that composed at least 0.1% of the reads in 25% of the samples. Heatmaps of relative abundances were constructed for all ASVs in all samples (Fig. S3); criteria for inclusion of ASVs in these heatmaps was a relative abundance within the top 10 most abundant ASVs in any sample within a respective dataset and greater

than 1% of the reads in any sample.

## 3 Results

### 3.1 Physicochemical conditions, inorganic nutrients, and chlorophyll-a

The mesocosm experiment occurred during an unusual warming event described as a coastal El Niño (Garreaud, 2018; Bach et al., this issue). Sea surface temperature (SST) was 1 to 1.5° C warmer than average (Herring et al., 2019) at

the start of the experiment. SST fluctuated over the experiment (Fig. 2a) but returned to average values about 5 weeks into the experiment. Temperature in the mesocosms tracked that of the surrounding water (Fig. 2a). Salinity in the mesocosms began with levels similar to the open ocean, and was modified via NaCl brine additions to depths below 10 m on day 13 and again on day 33 (Fig 2b).





**Figure 2.** Physicochemical conditions, inorganic nutrients, and chlorophyll-a (chl-a) in the mesocosms and unenclosed Pacific surface waters (a-e), with values averaged over the 0-17 m depth range. Flow cytometry results (f-j) from samples collected in the surface waters; these depths were 0 – 5 m on days 1 and 2, 0 – 10 m from day 3 to 28, and 0 – 12.5 m from day 29 to 50. For all panels, each individual sample is indicated. For (a), (b), (c), (d), and (e), the solid grey line represents the mean for all eight mesocosms; the shaded grey line is the mean ± SE for all eight mesocosms. The green lines indicate exchange of ~20 m³ of OMZ water with water in the mesocosm (total volume ~ 50 m³) on days 11 and 12. The orange lines indicate NaCl brine additions on days 13 (69 L at a depth range of 10 – 17 m) and 33 (46 L at a depth range of 12.5 – 17 m). (a) Temperature. (b) Salinity. (c) NO₂⁻+NO₃⁻. (d) Si(OH)₄. (e) Chlorophyll-a (chl-a). (f) *Synechococcus*. (g) Photosynthetic eukaryotes. (h) Cryptophytes. (i) Heterotrophic (non-pigmented) bacteria. (j) Small bacteria (smaller than standard non-pigmented bacteria in the system).




$NO_2^-$ + $NO_3^-$ ($NO_x^-$) concentration in M1 was initially 6.9 µmol $L^{-1}$ and declined steadily over time (Fig. 2c). M1 received addition of OMZ water with $NO_x^-$ concentration of 0.3 µmol $L^{-1}$ on days 11 and 12, and reached the detection threshold of 0.2 µmol $L^{-1}$ around day 20 (Fig. 2c). Conversely, $NO_x^-$ concentrations in the Pacific at the study site were considerably higher and more variable, ranging between 2.7 – 19.2 µmol $L^{-1}$ and reaching particularly high

values during the second half of the experiment (Fig. 2c). The concentration of $Si(OH)_4$ in M1 was initially 8.0 µmol $L^{-1}$ and declined to a value of 4.7 µmol $L^{-1}$ on day 10, prior to addition of OMZ water with $Si(OH)_4$ concentration of 17.4 µmol $L^{-1}$. The addition of OMZ water caused $Si(OH)_4$ in M1 to increase to 9.3 µmol $L^{-1}$ on day 13, after which its concentration declined for the rest of the experiment, reaching a value of 3.6 µmol $L^{-1}$ on day 50 (Fig. 2d). In the Pacific, $Si(OH)_4$ concentrations fluctuated, ranging from a minimum of 6.6 µmol $L^{-1}$ on day 24 to a maximum of 18.7

µmol $L^{-1}$ on day 12 (Fig. 2d).

Chlorophyll-a (chl-a) concentration in M1 was initially 4.9 µg $L^{-1}$ and declined until reaching a minimum of 1.7 µg $L^{-1}$ on day 6. Following OMZ water addition, chl-a gradually increased until approximately day 40, after which it increased rapidly. This rapid increase in chl-a in the final ~10 days of the experiment, which was seen across all eight

mesocosms (Fig. 2e), has been attributed to "orni-eutrophication" by defecating sea birds (Inca Terns, *Larosterna inca*), who found the mesocosms to be suitable resting places (Bach et al., this issue). Chl-a in the Pacific was more variable and generally comparable to the range of chl-a concentrations found in the mesocosms until day 40.

The observed values of temperature, salinity, $NO_x^-$, $Si(OH)_4$, and chl-a in M1 were very similar to the values found in

all eight mesocosms (Fig. 2), indicating that M1 was representative of the overall abiotic and hence biotic conditions found within the mesocosms.

### 3.2 eDNA overview statistics

Across the five datasets, the unenclosed Pacific samples exhibited higher alpha diversity relative to mesocosm at nearly all levels of taxonomy from phyla to ASVs. Summary statistics are given in Table 1. A total of 1,219,350 COI,

1,759,671 18S rRNA, 217,398 12S rRNA, 241,783 16S rRNA plastidial, and 3,125,836 16S rRNA bacterial paired end reads passed filtering and decontamination steps. These amplicons resulted in a total of 2,434 COI ASVs, 3,296 18S rRNA ASVs, 470 12S rRNA ASVs, 258 16S rRNA chloroplast ASVs, and 5,786 16S rRNA bacterial ASVs (Table 1). These resulting ASVs form the basis of the statistical analysis presented below.







**Table 1. DNA sequencing statistics. The 16 total samples were divided between 8 M1 (mesocosm) and 8 Pacific samples, with "Avg. reads" and "Stdev" reflecting the average and standard deviation (respectively) of the number of reads per sample. All other values represent the total sum across all samples. Across all five datasets, the Pacific samples consistently show a higher level of alpha diversity at all levels of taxonomy.**

|  | Cytochrome Oxidase I | | 18S rRNA | | 12S rRNA | | 16S rRNA Plastidial | | 16S rRNA Bacterial | |
| --- | --- | --- | --- | --- | --- | --- | --- | --- | --- | --- |
|  | M1 | Pacific | M1 | Pacific | M1 | Pacific | M1 | Pacific | M1 | Pacific |
| Avg. Reads | 78,140 | 74,279 | 125,809 | 94,150 | 13,236 | 13,939 | 9,131 | 21,092 | 203,083 | 187,646 |
| (Stdev) | 22,681 | 6,018 | 26,365 | 19,034 | 9,080 | 6,052 | 6,511 | 10,738 | 20,875 | 36,331 |
| ASVs | 1,720 | 1,890 | 2,105 | 2,825 | 360 | 265 | 158 | 203 | 3,223 | 4,139 |
| Phyla | 25 | 28 | 29 | 39 | 2 | 4 | 5 | 7 | 24 | 38 |
| Classes | 54 | 59 | 71 | 84 | 5 | 8 | 13 | 13 | 43 | 72 |
| Orders | 104 | 117 | 143 | 176 | 18 | 24 | 15 | 16 | 130 | 166 |
| Families | 141 | 168 | 193 | 229 | 25 | 32 | 14 | 15 | 183 | 242 |
| Genera | 45 | 57 | 214 | 242 | 24 | 27 | 2 | 4 | 348 | 414 |
| Species | 37 | 48 | 197 | 234 | 21 | 21 | 2 | 5 | 10 | 8 |

### 3.3 Results of Principal Component Analysis

#### 3.3.1 RPCA and PERMANOVA

The first principal component of COI, 18S rRNA, V1- V2 16S rRNA plastidial ASVs, and V1-V2 16S rRNA bacterial ASVs revealed a swift divergence between the Pacific and mesocosm communities (Fig. 3). In the COI, 18S rRNA, and 16S rRNA plastidial sequences (Fig. 3a, 3b, and Fig. 3c), this divergence is first seen on day 15, the first eDNA sample taken after the OMZ water and salt additions. In the 16S rRNA bacterial sequences (Fig. 3d), this divergence is already seen on day 8, prior to additions. A PERMANOVA test revealed a significant difference between M1 samples following additions and the other samples, confirming the significance of the observed divergence (COI: $p$ = 0.001, $F$-statistic = 11.78; 18S rRNA: $p$ = 0.001, $F$-statistic = 11.76;16S rRNA chloroplast: $p$ = 0.002, $F$-statistic = 7.20; 16S rRNA bacterial: $p$ = 0.001, $F$-statistic = 10.11). Following this initial divergence, the M1 samples continue to separate from the Pacific samples. The Pacific samples show a slight temporal trend on PC2 across all four datasets. As shown by the proportion of variance explained by each principal component, the degree of divergence between M1 and the Pacific is greater than the degree of change experienced by the Pacific.



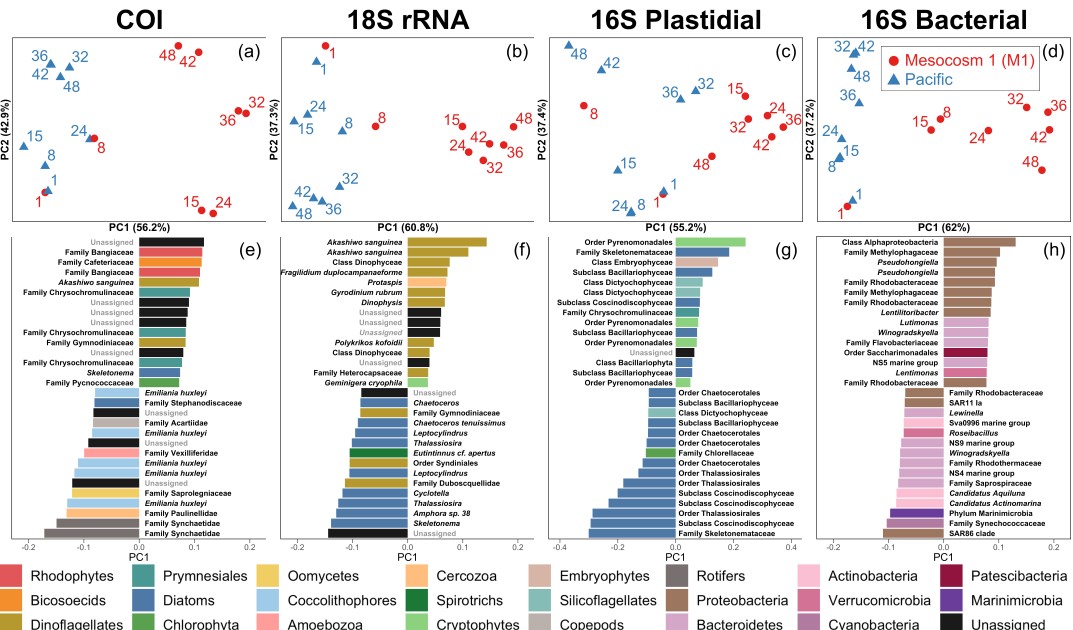

**Figure 3.** Beta diversity analyses using DEICODE robust Aitchison PCA for (a) COI, (b) 18S rRNA, (c) V1-V2 16S rRNA plastidial ASVs, and (d) V1-V2 16S rRNA bacterial ASVs, including photosynthetic taxa. Data points represent individual samples; the number next to each data point indicates the day collected. ASV loadings on PC1 from the DEICODE robust Aitchison PCA for (e) COI, (f) 18S rRNA, (g) V1-V2 16S rRNA plastidial ASVs, and (h) V1-V2 16S rRNA bacterial ASVs. As seen in (a), (b), (c), and (d) and confirmed by a PERMANOVA test, PC1 separates the Pacific samples and pre-OMZ water addition M1 samples from the post-OMZ water addition M1 samples (days 15+), and thus positive loading scores are associated with the mesocosm, whereas negative loading scores are associated with the Pacific samples. Text adjacent to loading scores represents the lowest taxonomy to which that ASV was annotated. The taxonomy assigned to 16S rRNA ASVs corresponds to that of the SILVA taxonomy.

### 3.3.2 Mesocosm-associated ASVs and taxonomy revealed by PC1 loading scores

The taxa that had the highest positive PC1 loading scores, and thus were associated with M1, included various photo-, mixo-, and heterotrophic taxa. For the COI data, this included two dinoflagellate ASVs, one assigned to *Akashiwo sanguinea* and the other to the family it belongs to, Gymnodiniaceae. Unassigned ASVs made up five of the other mesocosm-associated ASVs for COI, while other mesocosm-associated ASVs included two assigned to the red algae family Bangiaceae, one to the heterotrophic nanoflagellate family Cafeteriaceae, three to the haptophyte family Chrysochromulinaceae, one to the diatom *Skeletonema*, and one to the prasinophyte family Pycnococcaceae (Class V prasinophytes, Bachy et al., 2022).

The mesocosm-associated 18S rRNA ASVs were dominated by dinoflagellates, which were 9 of the top 15 positive loading ASVs. The six other ASVs in the top 15 mesocosm-associated ASVs included one ASV assigned to the cercozoan genus *Protaspis*, another ASV assigned to the cryptophyte species *Geminigera cryophila*, and four unassigned ASVs.



In the V1-V2 16S rRNA plastidial sequences, mesocosm-associated ASVs included four ASVs assigned to the cryptophyte order Pyrenomonadales, six diatom ASVs, an ASV assigned to the class Embryophyceae (land plants),

two ASVs assigned to the silicoflagellates (class Dictyophyceae), an ASV assigned to the family Chrysochromulinaceae, and an unassigned ASV. In the V1-V2 16S rRNA bacterial sequences, nine ASVs were assigned to the phylum Proteobacteria, four to the phylum Bacteroidetes, and one each to the phyla Patescibacteria and Verrucomicrobia.

### 3.3.3 Pacific ASVs and taxonomy revealed by PC1 loading scores

Pacific COI ASVs included three mesozooplankton ASVs, two assigned to the rotifer family Synchaetidae and one assigned to the calanoid copepod family Acartiidae. In the phytoplankton portion of the COI Pacific-associated ASVs, the coccolithophore *Emiliania huxleyi* figured prominently, comprising five of the top 15 Pacific-associated ASVs. The diatom family Stephanodiscaceae was also determined to be associated with the Pacific. Of the COI ASVs associated with the Pacific that weren't mesozooplankton or phytoplankton, one ASV belonged to the amoeboid

family Paulinellidae (phylum Cercozoa), one was assigned to the amoebozoan family Vexilliferidae, one was assigned to the oomycete family Saprolegniaceae, and three ASVs were unassigned.

Pacific 18S rRNA ASVs were dominated by diatom ASVs, which made up 9 of the top 15 Pacific-associated ASVs. All of these diatom ASVs were annotated to the genus- or species-level, and included the genera *Skeletonema*,

*Amphora*, *Thalassiosira*, *Cyclotella*, *Leptocylindrus*, and *Chaetoceros*. Three of the top 15 Pacific ASVs were dinoflagellate ASVs, and two of which were annotated to the order Syndiniales, with one of these ASVs annotated to the family Duboscquellidae within the Syndiniales; the other dinoflagellate ASV was annotated to the family Gymnodiniaceae. One other ASV was assigned to the tintinnid ciliate species *Eutintinnus* cf. *apertus*, and two other ASVs were unassigned.


The Pacific V1-V2 16S rRNA plastidial ASVs were dominated by diatoms, which represented 13 of the top 15 Pacific-associated ASVs. Of the two other Pacific-associated chloroplast-derived ASVs, one was assigned to the chlorophyte family Chlorellaceae and the other to the class Dictyophyceae (silicoflagellates). Among the 16S rRNA bacterial ASVs, the top 15 contained six assigned to the phylum Bacteroidetes, three each assigned to the phyla Proteobacteria

and Actinobacteria and one each assigned to the phyla Marinimicrobia, Verrucomicrobia and Cyanobacteria. Surprisingly, the cyanobacterial ASV is identical to a *Cyanobium* sp. strain, Suigetsu-CR5, isolated from a Japanese saline lake (Ohki et al., 2012), while further analysis of other, less abundant cyanobacterial ASVs identified more typical marine *Synechococcus* strains (see Supplement).

### 3.4 Vertebrate detection

12S rRNA detected a diversity of vertebrate taxa in the Pacific, dominated by anchovy (genus *Engraulis*) (Fig. 4a), and with fluctuating numbers of reads. In contrast, the mesocosm showed an initial high number of reads, similarly dominated by anchovy, followed by a drastic drop in reads in the second and third samples (taken on days 8 and 15),



which had only 10 and 37 vertebrate reads, respectively (Fig. 4b). The number of vertebrate reads in the mesocosm then increased on day 24 due to reads assigned to the family Sciaenidae (drums and croakers), and subsequently an

increase in anchovy reads was also observed. In Fig. 4c, reads assigned to the genus *Paralichthys* appear on days 32 and 36 following the addition of eggs of Fine Flounder *Paralichthys adspersus* on day 31, but are absent in all other samples. Other taxa of note that were detected by 12S rRNA but are not pictured in the above plots because they did not meet the abundance threshold to be included were Inca tern *Larosterna inca* (6 reads each in the mesocosm on days 42 and 48) and brown pelican *Pelecanus occidentalis* (159 reads in the mesocosm on day 36); the detection of

these species is indicated by the corresponding symbols above the bars in Fig. 4b.

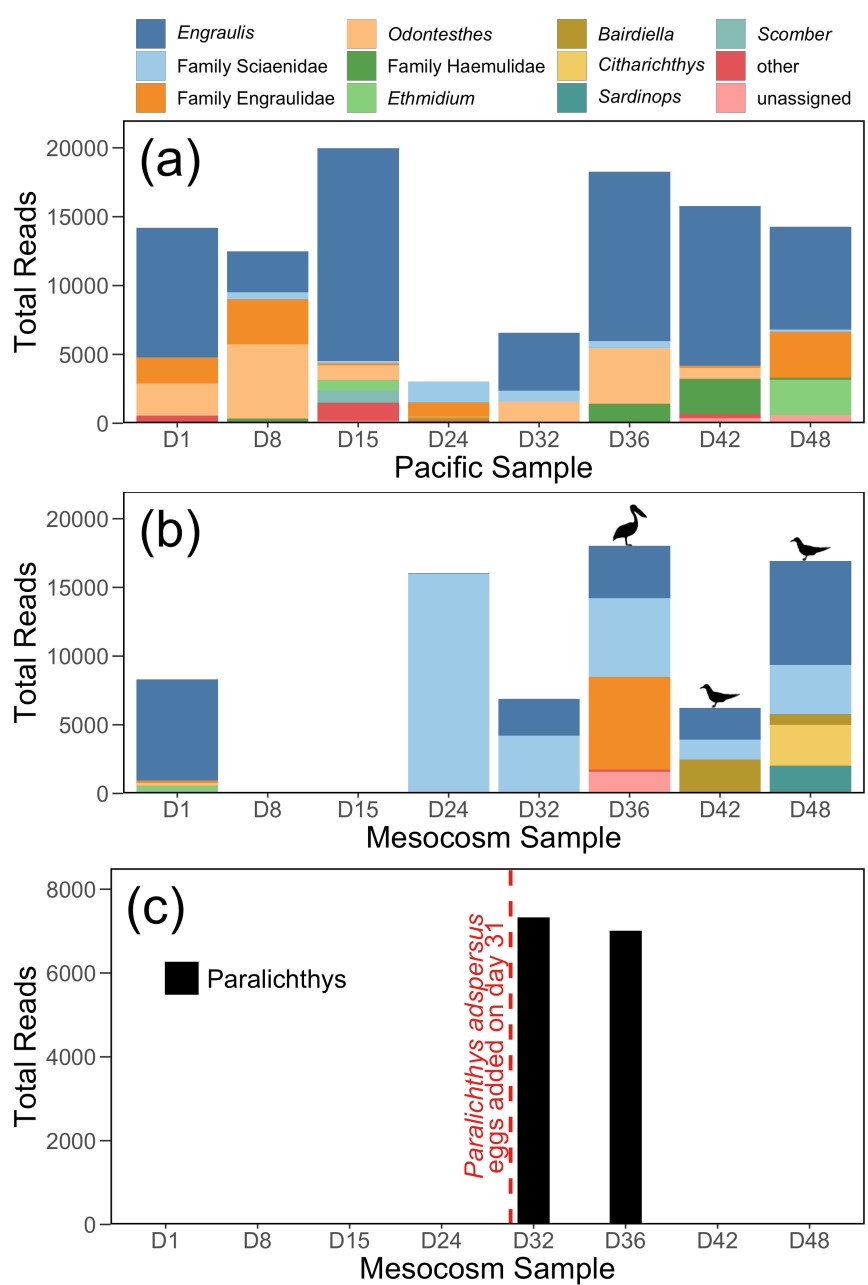

**Figure 4. 12S rRNA results, showing top 10 genera, (or families, for ASVs that were not annotated to the genus level) for (a) the Pacific and (b) the mesocosm samples, by total number of reads. Detections of *Larosterna inca* and *Pelecanus occidentalis* are indicated by the symbols in (b). All reads of *Paralichthys* have been removed from (a) and (b). (c) shows the total number of reads of *Paralichthys*, the genus of the fish eggs (*Paralichthys adspersus*) that were added to the mesocosm on day 31. Because there is no available reference sequence for *Paralichthys adspersus*, genus-level annotations are used to assess the detections of this species using eDNA.**




## 4 Discussion

The Peruvian EBUS is known for its dynamic nature (Penven et al., 2005; Huyer et al., 1991; Echevin et al., 2004),
reflected by the physicochemical (warming shift, $NO_x^-$ fluctuations) and ecological (Fig. 3) changes observed in the
unenclosed Pacific ocean over our 50-day sampling period. The variability observed in the Pacific during the sampling
period is contrasted by the large shift from these upwelling conditions in the mesocosm, which was isolated from the
surrounding Pacific and stratified by NaCl brine injections. This resulted in reduced horizontal and vertical mixing,
leading to depletion of nutrients (Fig. 2); $NO_x^-$ (Fig. 2c) in particular exhibited a quick and continuous decline, reaching
the threshold of detection by day 20. This is in stark contrast to the concentration of $NO_x^-$ in the Pacific, which varied
from 2.7 – 19.2 µmol L$^{-1}$. The observed high concentrations and high day-to-day variability of nitrogen in the natural
Pacific samples was indicative of intermittent upwelling, whereby nutrients in the surface waters are replenished from
below (Chavez and Messie, 2009; Graco et al., 2017). $Si(OH)_4$ (Fig. 2d) is also lower in the mesocosm and declined
in concentration throughout (except for when OMZ water, rich in $Si(OH)_4$, was injected), but never reached levels
below 3.4 uM.

The differences in physical and biogeochemical conditions between the Pacific and the eight mesocosms were
pronounced. These physical and chemical differences are reflected in the eDNA metabarcoding comparisons of
diversity and taxonomic composition of the biological communities in the unenclosed Pacific Ocean and one of the
mesocosms (M1). The communities responded in a somewhat predictable manner, with those in the Pacific being
characteristic of higher nutrient, upwelling conditions and those in the mesocosms being characteristic of lower
nutrient, stratified conditions.

### 4.1 Phytoplankton communities

Phytoplankton are strongly linked to changes in abiotic conditions in upwelling systems, and the connection between
environmental conditions and phytoplankton was reflected in the differing communities of the mesocosm and Pacific.
Upwelling conditions, as were observed in the Pacific during the experiment, are associated with increased nutrient
availability and turbulence (Chavez and Messié, 2009), favouring fast growing non-motile phytoplankton such as
diatoms (Margalef, 1978), and newly discovered prasinophyte species (see e.g., Worden et al., 2004; Simmons et al.,
2016). Conversely, relaxing upwelling conditions and subsequent stratification (as were observed in the mesocosm)
result in reduced mixing, and exhaustion of nutrients (nitrate) in the surface layer. These conditions favour motile
organisms like dinoflagellates who are able to vertically migrate to the deeper nutrient-rich layer at night and back to
the surface during the day without the hindrance of turbulence (Margalef, 1978; Smayda, 2010; Smayda and Trainer,
2010).

The stratified mesocosm was dominated by the mixotrophic dinoflagellate *Akashiwo sanguinea*, a signature bloom
species in EBUS (Trainer et al., 2010; Bach et al., this issue). *A. sanguinea* ASVs were found to be strongly mesocosm-
associated in both the COI and 18S rRNA datasets (Fig. 4e and 4f), and in the 18S rRNA sequences in particular, *A.
sanguinea* was dominant in the mesocosm in the later stages of the experiment, comprising over 50% of the reads in



some samples (Fig. S3). *A. sanguinea* reads were a minor proportion of the reads in the COI dataset and absent from

the 16S rRNA plastid sequences; the disparity in the detection of *A. sanguinea* by our different PCR primers can be explained by inefficient dinoflagellate amplification by COI primers (Lin et al., 2009) and the loss of most chloroplast genes from dinoflagellates (Koumandou et al., 2004), causing them not to be detected in our 16S rRNA plastidial sequences (Needham and Fuhrman, 2016). In this experiment, *A. sanguinea* was also identified as the dominant dinoflagellate by imaging flow cytometry and microscopy (Bach et al., this issue). Bach et al. (this issue) report the

common diurnal migration pattern of dinoflagellates, at the surface during the day, migrating to the nutricline at night to take up nutrients. Mixotrophy and vertical migration allow *A. sanguinea* to exploit nutrients found below the nutrient-depleted surface layer, especially when there is a shallow thermocline (Kudela et al., 2010), as was the case in the mesocosm in this experiment (Fig. 3a in Bach et al., this issue). Blooms of *A. sanguinea* are regularly observed in EBUS (Du et al., 2011; Kudela et al., 2008; Kahru et al., 2004; Dugdale et al., 1977) under shallow thermocline,

stratified conditions, following upwelling relaxation. *A. sanguinea* is typically not found in EBUS phytoplankton communities outside these specific conditions (Kolody et al., 2019; Limardo et al., 2017). Hence, the mesocosm experiment provided the right environmental conditions to initiate a dinoflagellate bloom from the propagules inside the mesocosm, which were either already present in the enclosed water or introduced from the OMZ water additions.

In addition to *A sanguinea*, many other dinoflagellate ASVs were found to be mesocosm-associated. These dinoflagellate ASVs included the mixotrophic dinoflagellate *Fragilidium duplocampanaeforme* and its prey, the genus *Dinophysis* (Park and Kim, 2010). A heterotrophic dinoflagellate known to feed on bloom-forming dinoflagellates, *Polykrikos kofoidii* (Matsuyama et al., 1999; Tillmann, 2004), was also detected to be mesocosm-associated. Dinoflagellates comprised about two-thirds of the top organisms in our 18S rRNA PC loadings analysis (Fig. 3f)

indicating a strong selection of these organisms under stratified conditions and interesting predator-prey relationships. Other mesocosm-associated ASVs included ones annotated to the family Chrysochromulinaceae in the COI and V1-V2 16S rRNA plastidial datasets, which also likely reflect stratification, as blooms of *Chrysochromulina* have previously been reported in strongly stratified, stable conditions (Edvarsen and Paasche, 1998). Multiple ASVs assigned to the cryptophyte order Pyrenomonadales were found to be mesocosm-associated in the 18S rRNA and V1-

V2 16S rRNA plastidial datasets; this could be connected to their green light harvesting phycobiliproteins, which appear to facilitate their growth in low-light conditions (Spear-Bernstein and Miller, 1989), as were found in the mesocosms (Fig. 3c in Bach et al., this issue). However, some Pyrenomondales are predatory mixotrophs, thus changes in relative abundance could also connect to changes in the bacterial community that provisioned prey resources.

In contrast, the phytoplankton community of the coastal Pacific ocean was dominated by diatoms, typical of communities during higher nutrient upwelling conditions (Pennington et al., 2006; Chavez et al., 2017; Choi et al., 2020) as well as the cosmopolitan coccolithophore *Emiliania huxleyi*. As seen by the high levels of $NO_x^-$ in the Pacific (Fig. 2c), upwelling conditions were maintained during the period of study, favouring *r*-selected species like diatoms and maintaining an early succession community (Margalef, 1978; Tyrrell and Merico, 2004). The presence of

coccolithophores is indicative of the mixing of some open ocean waters that had come closer to shore due to the coastal





El Niño. Although Class II prasinophytes (Mamiellophyceae) were detected in the 16S amplicon data (Fig. S3), they were present at lower relative abundances than the above groups. This is different from observations of prasinophyte importance in more mesotrophic water columns in EBUS off California, USA (Limardo et al., 2017; Kolody et al., 2019; Choi et al., 2020) and off the coast of Chile (De la Iglesia et al., 2020). In the mesocosm, the low levels of

nutrients in the mixed layer resulted in lower diatom growth rates and they sank due to increased stratification (Smayda and Trainer, 2010). Some diatom ASVs remained prevalent in the mesocosm, particularly ASVs assigned to the family Skeletonemataceae (Fig. S3), perhaps as seed populations ready to exploit a change in the environment. In the latter part of the experiment they may also have been seeded by the seabird faeces, given the increased presence of birds in the local area. Diatoms are ubiquitous in anchovy stomachs (Espinoza and Bertrand, 2008) and anchovies are favoured

by seabirds (Duffy, 1983), and both anchovy and seabird eDNA appeared later in the mesocosm.

There were also three dinoflagellate ASVs common in the Pacific. While one of these was assigned to the family Gymnodiniaceae, a broad family of free-living dinoflagellates, the other two were both assigned to the Syndiniales, a parasitic order (Guillou et al., 2008). Among these two ASVs, one was annotated to the family Duboscquellidae, with

some species within this family known to parasitize the tintinnid ciliate *Eutintinnus* (Coats, 1988). The ciliate *Eutintinnus* has also been documented to form symbiotic relationships with multiple diatom species (Gómez, 2007, 2020; Vincent et al., 2018). Interestingly, an ASV assigned to the genus *Eutintinnus* was also found in the Pacific, suggesting that both the parasite-host relationship and the symbiotic relationship of this ciliate was found in the ocean surrounding the mesocosms.


An unusual and abundant ASV identical to *Cyanobium* sp. Suigetsu-CR5 (Ohki et al., 2012) was found in the coastal Pacific samples. The primers used to amplify 16S rRNA did not provide full-length 16S rRNA gene sequences to confirm that this ASV was the same organism as that isolated from a Japanese lake. However, *Cyanobium* sp. Suigetsu-CR5–like sequences have been reported from the open ocean Northeastern Pacific (Sudek et al., 2015). Taken together,

these results suggest that a marine *Cyanobium* sp. Suigetsu-CR5–like organism exists and given the right environmental conditions can become a dominant cyanobacterium.

### 4.2 Zooplankton communities

The mesozooplankton communities of the mesocosm and Pacific as detected via COI were differentiated primarily by Pacific-associated ASVs assigned to the rotifer family Synchaetidae and the calanoid copepod family Acartiidae (Fig.

3e); these results are in agreement with vertical tows of an Apstein net of mesh size 100 µm during the experiment (unpublished data). Rotifers were detected initially in similar abundance in both the mesocosm and Pacific in the COI data, but declined in abundance in the mesocosm while remaining more abundant, albeit highly variable, in the Pacific (Fig. S3b). Conversely, ASVs assigned to the copepod family Acartiidae were more abundant in the Pacific than the mesocosm from the start of the experiment in both the net tow and COI data (unpublished data, Fig. S3b), indicating

that fewer individuals of this family were trapped within the mesocosm when it was closed, owing to the patchy distribution of zooplankton (Wiebe and Holland, 1968). The most dominant zooplankton taxa found were the calanoid





copepod *Paracalanus* (detected by COI and 18S) and the cyclopoid copepod *Hemicyclops* (detected only by 18S); however, these taxa were found in similar abundances in both the mesocosm and Pacific samples throughout the experiment by both eDNA metabarcoding and net tow data and thus did not differentiate the two sampling sites (Fig.

S3b and S3c). Copepods have lifespans that are greater than the duration of the experiment (Ianora, 1998), so a significant response of the copepod communities was not captured within the mesocosm relative to the Pacific. Longer term experiments will be required to study responses of animals with generation times of months or greater.

The nanozooplankton community of the mesocosm was separated from that of the Pacific by the higher relative
abundance of the family Cafeteriaceae, the higher relative abundance of the cercozoan genus *Protaspis*, and the relative lower abundance of the families Paulinellidae and Vexilliferidae. The family Cafeteriaceae is a made up of heterotrophic nanoflagellates, many of which are filter-feeding bacterivores (Schoenle et al., 2020). Their increased abundance in the mesocosm may reflect their tendency to associate with detritus (Patterson et al., 1993) or adhere to structure (Baker et al., 2018; Boenigk and Arndt, 2000), as was available in the mesocosm. The amoebozoan family
Vexilliferidae is ubiquitous in both marine and estuarine environments (Page, 1983), while the amoeboid family Paulinellidae (phylum Cercozoa) contains phototrophic and heterotrophic species which inhabit freshwater, brackish, and marine environments (Kim and Park, 2016).

### 4.3 Heterotrophic and photosynthetic bacterial community

As seen in the beta diversity analyses in Fig. 3, the bacterial community of the mesocosm diverged from that of the
Pacific faster than the communities detected by COI, 18S rRNA, and 16S plastidial sequences, separating on PC1 by day 8 rather than day 15. This may be at least partially explained by the fast growth rate of prokaryotic cells (Zubkov, 2014). The ASVs that were Pacific-associated were representative of typical coastal, phytoplankton rich communities (Needham and Fuhrman, 2016; Buchan et al., 2014). These ASVs included an ASV assigned to the 16S rRNA clade SAR86, one of the most abundant constituents of microbial communities in the surface ocean (Dupont et al., 2012),
and an ASV assigned to the family Synechococcaceae, a ubiquitous family of cyanobacteria that is most abundant in nutrient-rich surface waters (Partensky et al., 1999), as was found in the Pacific during the course of the experiment (Fig. 2). Mesocosm-associated ASVs included the methylotrophic family Methylophagaceae (Neufeld et al., 2007), *Pseudohongiella*, a genus recently isolated from the northwest Pacific (Xu et al., 2016; Park et al., 2014), multiple ASVs of the family Rhodobacteraceae, and an ASV assigned to the NS5 marine group. Rhodobacteraceae have
previously been documented to increase in response to a bloom of *A. sanguinea* (Yang et al., 2012), as was documented within the mesocosm, and have also been documented to increase in blooms of the harmful dinoflagellate *Alexandrium* (Hattenrath-Lehmann and Gobler, 2017). Similarly, the NS5 marine group has previously been documented to have increased in abundance with blooms of both *A. sanguinea* (Yang et al., 2015) and *Alexandrium* (Hattenrath-Lehmann and Gobler, 2017).


One of the most unusual taxa that was detected was from the order Saccharimonadales within the superphylum *Cand.* Patescibacteria (Parks et al., 2018), which was first detected in M1 on day 15 and in the Pacific on day 24, and



subsequently increased in abundance in both M1 and the Pacific. The rise corresponded with an increase in small bacteria and heterotrophic bacteria overall in flow cytometry samples. While this group (Saccharibacteria) has been
detected in seawater previously (Hugenholtz et al., 2001) and is commonly found across many different environments from soil to the human gut and oral microbiomes (Kuehbacher et al., 2008; Marcy et al., 2007; Ferrari et al., 2005), some representatives of this group have an unusual symbiotic lifestyle. They were recently isolated from the human oral microbiome where they were found to be epibionts of Actinobacteria (He et al., 2015). Recent genomic evidence reveals that Saccharibacteria have small genomes (< 1 Mb) with genomic contents consistent with a symbiotic lifestyle
(Lemos et al., 2019), and microscopy and filter size-fractionation have found them to be very small cells (or "ultra-small bacteria"). To account for their rapid rise in prevalence, it is possible that these Saccharibacteria colonized the mesocosms via a biofilm (outside and inside the experiment itself) or were introduced via seabird faeces, as Saccharibacteria have previously been detected in the avian microbiome (Hird et al., 2015). The timing of the increase is consistent with the appearance of the seabirds on the mesocosms. Since the seabirds were foraging in the vicinity
of the mesocosms it is possible that they may have contributed to the increase in the Pacific samples as well. In any case, this observation warrants further study to understand the lifestyle and genomic potential of this enigmatic group.

### 4.4 Insights into vertebrates

The 12S rRNA metabarcoding data detected distinctly different communities in the Pacific and in the mesocosm. Bony fishes typical of the coastal Peruvian upwelling system, dominated by the Peruvian anchoveta (*Engraulis*
*ringens*) were found in the Pacific samples. In the mesocosm, the species assemblage reflected three experimental perturbations: 1) the exclusion of vertebrates via a 3 mm mesh, 2) the addition of eggs of *Paralichthys adspersus*, and 3) the impact of resting seabirds. During the deployment of the mesocosm, the initial water was filtered through a net of mesh size 3mm, and collection of the OMZ water that was later added was conducted through a net of mesh size 10mm. The fine mesh sizes effectively excluded most, if not all, living vertebrate stages from the enclosure but not
their eDNA. The initial vertebrate eDNA in the mesocosm decayed rapidly as evidenced by the extremely low read counts on days 8 and 15 (10 and 37 vertebrate reads, respectively). The virtual disappearance of vertebrate eDNA a week after the start of the experiment is in line with estimates of bony fish eDNA decay rates (Sassoubre et al., 2016), which show an exponential decline in eDNA concentrations with time. Quantification of eDNA decay rates of northern anchovy (*Engraulis mordax*), which is congeneric with the Peruvian anchoveta *E. ringens*, showed that *E. mordax*
DNA concentration reached the threshold of detection by a qPCR assay within three days of the fish being removed from the enclosure (Sassoubre et al., 2016). There may have been some eDNA introduced by the addition of the OMZ water; however, the ten-day period between when this water was collected on day 5 and when the first eDNA sampling occurred following OMZ water addition (day 15) coupled with the decay rate of eDNA makes the detection of this eDNA unlikely.


The addition of eggs of the fish *Paralicthys adspersus* ("Fine Flounder") to the mesocosm on day 31 led to a notable spike in *Paralichthys* reads on days 32 and 36; *Paralichthys* was not detected in any other samples in the mesocosm or Pacific. Because *P. adspersus* does not have a reference sequence available in GenBank (release 238.0) for the





region targeted by the MiFish primers, no species-level detections were possible, and thus we assessed the detection of this species by eDNA through annotations to the genus *Paralichthys*. The strong signal of *Paralichthys* in the 12S rRNA results is particularly notable because the introduction of fish eggs was not detected via net sampling (Bach et al., this issue). The fish eggs apparently did not develop and decomposed or sank out of the mesocosm fairly quickly. Based on their size, the estimated sinking rate for *P. adspersus* eggs, which average a diameter of 0.66-0.80 mm (Silva and Oliva, 2010), is about 5 mm s$^{-1}$ (Robertson, 1981). Additionally, for any eggs that hatched, larval settling rates for

two other species of *Paralichthys* is estimated to be about 10 mm s$^{-1}$ (Hare et al., 2006). With the very stable, stratified waters of the mesocosm, the lack of turbulence likely would have caused any eggs or larvae to quickly settle out of the water column, and coupled with eDNA decay rates, it is not surprising that this signal was no longer detected in the eDNA sample taken on day 42.

The final and most pronounced influence on the vertebrate composition within the mesocosm was the impact of seabirds, which occasionally rested on the mesocosms until day 36, but increased abruptly in abundance thereafter (Bach et al., this issue). DNA from both seabirds and their prey was detected in the second half of the experiment. Inca tern (*Larosterna inca*), the most common bird observed on the mesocosms, was detected in the last two mesocosm samples (days 42 and 48), while brown pelican (*Pelecanus occidentalis*) was detected in the mesocosm on day 36

(Fig. 4b). While bony fish are the intended target of the MiFish primers, the seabird sequence similarity as vertebrates resulted in weak amplification; thus, while we detected them, they are likely not well represented in our data. Primers that specifically target birds should improve assessments of their eDNA (Ushio et al., 2018). Inca terns are known to feed primarily on *E. ringens*, and the defecation by these seabirds into the mesocosm is inferred by the increase in *Engraulis* DNA in the last four samples. Additionally, many reads assigned to the family Sciaenidae (drums and

croakers) were detected in the mesocosm, but because of the lack of reference sequences for most of the species in this family that are found in the coastal waters of Peru, it is impossible to determine the identity of these species. However, members of the family Sciaenidae have been documented to be important prey species for seabirds in other regions (Lamb et al., 2017). In addition, the increase in small bacteria (Fig. 2j) in both the mesocosm and the Pacific may be a result of the seabirds defecating in and around the mesocosm.

**4.5 Strengths and limitations of eDNA metabarcoding in mesocosm experiments**

Through the use of eDNA metabarcoding targeting multiple genetic loci, we were able to examine community-level changes, detecting 12,244 ASVs assigned to 85 unique phyla; amongst these ASVs, the taxonomy of 816 ASVs could be resolved to the genus level. Sampling methods commonly used to assess communities often inherently size-select (vertical net tows, Skjoldal et al., 2013), are hindered by both the labour-intensive nature of morphological assessments

and the inherent variation in expertise amongst taxonomists (Harvey et al., 2017), and are limited in the taxonomic resolution they provide (e.g. flow cytometry). eDNA metabarcoding provides a method that overcomes some of these limitations: by sampling genetic material in seawater, there is higher sensitivity for detection, no minimum size threshold (as there is in light microscopy or net tows), and metabarcoding of genetic material provides objective taxonomic assignments at higher resolution than morphological identification (Berry et al., 2015), depending on the





variable region sequenced and its efficacy for resolving different taxonomic groups (Wear et al., 2018; Parada et al., 2016).

eDNA metabarcoding also comes with a number of challenges and limitations, particularly the issue of incomplete reference databases. For example, amongst the 88 COI ASVs that met the criteria for inclusion in the analysis of PC1

loading scores (Fig. 3e-h), only 17 had a match greater than 95% to a reference sequence. Most of the COI ASVs identified as driving differences between the mesocosm and Pacific samples and visualized in Fig. 3e had percent matches to reference sequences of 80-90% and thus were either annotated to higher orders of taxonomy or left unassigned; the only exceptions to this are the ASVs assigned to *Akashiwo sanguinea*, *Emiliania huxleyi*, *Skeletonema*, and Class V prasinophytes. A similar issue occurred with the 12S rRNA data – because of the lack of reference

sequences for Peruvian fishes, most ASVs could only be annotated to the genus or family level. While reference databases are constantly improving thanks to large-scale sequencing efforts (e.g., Rimet et al., 2016; Gold et al., 2021), the lack of reference sequences will continue to hinder metabarcoding efforts, especially for uncharismatic and cryptic taxa.

### 4.6 Summary and conclusion

In this study multiple marker eDNA metabarcodes were used to follow the evolution of diverse communities in an *in situ* mesocosm and the nearby unenclosed Pacific Ocean over 50 days. A quick divergence between mesocosm and open ocean communities was observed at multiple trophic levels. The mesocosm community evolved quickly into one that is common in stable, stratified conditions (Smayda and Trainer, 2010; Margalef, 1978). The Pacific community retained the character of communities found under nutrient-replete, upwelling conditions. Upwelling conditions are

associated with increased nutrient availability and turbulence (Chavez and Messié, 2009), favouring fast-growing, non-motile phytoplankton such as diatoms (Margalef, 1978). Conversely, weak upwelling and increased stratification result in reduced mixing and removal of nutrients from the shallow mixed layer (Smayda and Trainer, 2010), favouring motile organisms like dinoflagellates who are able to migrate to the nutricline at night without the hindrance of turbulence (Smayda, 2010; Margalef, 1978). The bacterial communities reflected the changes in phytoplankton

composition and likely the influence of seabirds. Many of the abundant bacterial taxa in the mesocosm are reported to increase in response to blooms of dinoflagellates, while typical open-ocean cyanobacteria (Synechococcaceae) and heterotrophic bacteria (e.g., SAR86) were more abundant in the Pacific. The increase in resting seabirds towards the end of the experiment led to unusual bacterial groups that have been reported to be present in the guts of animals (namely the Saccharibacteria). Primary and secondary consumers in upwelling ecosystems, such as zooplankton and

fish, have slower responses to physicochemical conditions so their responses were incompletely resolved in the 50-day experiment.

The method of eDNA metabarcoding is rapidly evolving, facing a series of challenges and opportunities. For example, incomplete reference database issues hindered the confidence of our taxonomic assignments. This was particularly

notable off the coast of Peru, where few species have been well-characterized genetically. Nonetheless, the multiple-



marker eDNA metabarcoding results presented here show how single samples can yield results that are comparable and complementary to a multitude of traditional and other emerging methods to survey marine biodiversity across the tree of life. The application of multiple genetic markers provided insight into how multiple trophic levels interact under changing physical and biological (seabirds) conditions, revealing coupled changes in bacterial (16S),

phytoplankton (18S), zooplankton (COI), and vertebrate (12S) communities. They also revealed evidence of potential predator-prey and parasite-host relationships, whose complexity could be explored further in interaction networks.

The effects of perturbations, either purposeful (additions of OMZ water, brine solution, and fish eggs) or unintended (seabird droppings), on marine communities were clearly resolved with our methods. The perturbations provided new

insights into ecosystem processes that are difficult to study otherwise. Mesocosm experiments are challenging because of the difficulties of reproducing physical conditions in contained systems. However, fundamental ecosystem processes in the ocean (i.e. stratification) can be well studied with mesocosms as described here. The selection of dinoflagellates under the warm (coastal El Niño) and stratified conditions in the mesocosm may be an indication of how EBUS will respond under the global environmental changes (i.e. continued global warming) forecast for the

future (IPCC, 2014). In support of this, evidence from the fossil record indicates that during the Paleocene thermal maximum, when temperatures where warmer than present, there was a global response of surface-dwelling coastal dinoflagellate communities (Crouch et al., 2001). Mesocosm experiments, like the one studied here, provide a valuable complement to traditional observational studies of upwelling systems. Insights from the genetic methods applied here will guide us towards a more mechanistic understanding of the processes that are shaping EBUS communities in a

changing ocean.

**Author contributions**

MM, DMN, SS, NKT, KP, AZW, and FPC wrote the paper. MM and DMN conducted the data analysis. GMC led the collection and preservation of eDNA samples with assistance from DMN and CP. CP led the collection and

preservation of flow cytometry samples with assistance from GMC and DMN. BG analysed the flow cytometry data. EE analysed the nutrients. UR and AL designed the overall mesocosm experiment.  FPC and AZW designed the sampling/analyses of samples described in the paper and secured funding to carry out the work.

**Competing interests**

The authors declare that they have no conflict of interest.

**Special issue statement**

This article is part of the special issue "Ecological and biogeochemical functioning of the coastal upwelling system off Peru: an in situ mesocosm study". It is not associated with a conference.



**Acknowledgements**

We thank the members of the GEOMAR team and Carlos Robles from IMARPE for providing logistical, physical, and moral support during the experiment. Funding was provided by the David and Lucile Packard Foundation through MBARI and NASA grant NNX14AP62A "National Marine Sanctuaries as Sentinel Sites for a Demonstration Marine Biodiversity Observation Network" to FPC, and the Gordon and Betty Moore Foundation, grant GBMF 3788 and NSF

DEB-1639033 to AZW.

**Code and data availability**

All code and data required to reproduce the analytical results, figures, and tables for this study are available on GitHub at https://github.com/MBARI-BOG/KOSMOS_eDNA_paper (last access: 26 October 2022;

https://doi.org/10.5281/zenodo.7255826, Min, 2022)

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
