# Peer review of "Ecological divergence of a mesocosm in an eastern boundary upwelling system assessed with multi-marker environmental DNA metabarcoding"

_Biogeosciences, 2022_

## Author Response (AR1)

**Associate Editor decision: Publish subject to minor revisions (review by editor)**
by Hans-Peter Grossart
**Public justification (visible to the public if the article is accepted and published)**:
The authors did a good job to address all concerns and suggestions by the reviewers. Of course, there remain some limitations such as the lack of replicates, which at this point can't be changed anymore. Of course, this limitation has to be clearly stated. Yet, the manuscript holds enough novelty and interesting results to be published.

**Additional private note (visible to authors and reviewers only):**
Dear authors, as stated above: You did a good job to address all concerns and suggestions by the reviewers. Of course, there remain some limitations such as the lack of replicates, which at this point can't be changed anymore. Of course, this limitation has to be clearly stated. There were some other minor issues which have been only partly addressed in your revision. Such as your response to the last comment of reviewer #2. Thus, I request that you carefully address these issues and revise your manuscript accordingly. Many thanks and best wishes, yours Hans-Peter

*Dear Dr. Grossart,*

*Thank you very much for handling our manuscript submission. The issue of replicates has been addressed in more detail in our revised manuscript on lines133-135 and 179-180. The other reviewer comments have now been addressed in full and the changes are included in the new version of the manuscript. The last comment by reviewer #2 regarding our figures has also now been addressed with revised figures included in the attached manuscript. These figures now have no bold text (to ensure that the font is uniform) and all axis text has been standardized in the code used to generate the figures. Below we put the authors comments in* **bold** *followed by our responses in italics. Please note that the line numbers listed in our responses correspond to the line numbers in the tracked changes file.*

*Best,*

*Markus*

**Comment on bg-2022-212**

Anonymous Referee #1

Referee comment on "Ecological divergence of a mesocosm in an eastern boundary upwelling system assessed with multi-marker environmental DNA metabarcoding" by Markus A. Min et al., Biogeosciences Discuss., https://doi.org/10.5194/bg-2022-212-RC1, 2022

**Min et al. performed a 50-day in situ mesocosm experiment in the Peruvian upwelling system to characterize ecological shifts in marine communities in response to upwelling events using macronutrient-rich seawater collected from an oxygen minimum zone. Their approach included using a suite of taxonomic biomarkers to capture genetic signatures across trophic levels, from bacteria to vertebrates. The initial goal seemed to be to compare simulated upwelling conditions (mesocosms) to non-upwelling conditions outside the**

enclosures. Instead, there was intermittent upwelling happening outside the enclosures, and the mesocosms themselves were quickly nutrient depleted and purposely stratified, characteristic of relaxed upwelling. The resulting communities in the mesocosm were dinoflagellate-dominated, which are known to dominate under stratified, low nutrient conditions.

The mesocosm experiment was an interesting exercise that seemed to evolve as the study went on. After the experiment began, the authors injected salt brine to the mesocosms to prevent vertical mixing and force stratification, added additional OMZ water, and also then added zooplankton larvae (although it's not immediately clear why the larvae were added). More details as to why these decisions were made would be helpful for readers. After 40 days, Inca terns miraculously figured out where they could safely rest on the enclosures, made a mess and stimulated their own phytoplankton bloom(!), with their influence observed in eDNA fraction in the final days of the experiment. These factors heavily influenced the physicochemical and nutrient conditions of the mesocosm and make it a more complicated comparison that originally planned to address. The authors do a commendable job interpreting results as a function of these factors. Caution is needed in extrapolating their results to how microbial communities in this ecosystem will respond under climate-induced scenarios given these multiple variables at play. Also, temperature does not appear different between the mesocosm and Pacific samples in Fig. 2. I recommend the authors tone down this language in the abstract.

*Thank you very much for your helpful comments. To provide more context for the manipulations of the mesocosms, we have added text to the methods in the paragraph on lines 147-162. This new text further describes the rationale for adding OMZ water, salt brine, and larvae. Regarding the comment on the addition of larvae: As part of the mesocosm experiment the group at large decided it would be interesting to study the responses of endemic organisms to the experimental conditions in the mesocosms, and therefore two endemic organisms, larvae of the Peruvian Scallop (Argopecten purpuratus) and eggs of the Fine Flounder (Paralichthys adspersus) were added to all mesocosms. This explanation can now be found on lines 155-158.*

*The reviewer also urges caution regarding extrapolating the local stratification results to larger climate warming scales, noting that surface temperatures were not different between the mesocosm and open ocean Pacific. We note that in the mesocosm stratification was driven by salinity, not temperature. Both salinity and temperature driven stratification have been implicated in dinoflagellate blooms in the coastal ocean (see Margalef, 1976 and others). The extrapolation of our results to future climate scenarios has been more carefully worded, both in the abstract (lines 42-45) and in the discussion (lines 774-778). In particular, on line 42 in the abstract, line 500 in the discussion, and line 775 in the conclusion, the fact that salinity, not temperature, was the primary driver of stratification in the mesocosm is noted.*

Specific Comments

Can the authors comment on why DNA samples were only taken from M1? Replication from multiple enclosures would have provided an assessment of variability and confidence in the resulting community following a period of relaxed upwelling. Did Akashiwo

**dominate in all other enclosures based on non-DNA evidence? I understand this information may be included in Bach et al., but would be useful to mention in this article as well for contextualization of their eDNA results.**

*Replication would have been very useful to improve confidence in our conclusions, but unfortunately DNA samples from only one mesocosm were analyzed because of resource constraints. This information has been added on lines 133-135, and it is made more explicit that there were no replicates on lines 178-179. We also note on lines 538-540 that based on observations using other sampling methods, Akashiwo dominated seven of eight mesocosms. These additions would help underscore that this metabarcoding result would have likely held in most other mesocosms, had they been analyzed.*

**The sequencing of blank filters was valuable and will be informative to others conducting amplicon sequencing on environmental samples. Can the authors speculate on where the diatom field-contamination came from? Gloves? Filtration rigs? Air?**

*It is impossible to say what the origin of field contamination is with any degree of certainty. We note that we followed clean protocols as best as possible in a very heavily used laboratory area. This level of contamination is not that different to what we find under more controlled experimental conditions. As such, we note on lines 202-205 that there was some field-based cross-contamination but choose not to speculate on its origin.*

**Line 272 – "The MEGAN6 parameter LCA percent was from 0.80 to 0.85, depending on the marker, allowing for up to 15 – 20% of top hits to be off target and still have the majority taxonomy assigned. This parameter value was chosen to allow for minor numbers of incorrectly annotated GenBank entries – effectively allowing for ASVs which had many high quality hits to a taxa to still be assigned to that taxa even if there existed a high-bitscore hit to another GenBank sequence annotated to an unrelated taxa."**

**If I am following correctly, an ASV could have been a strong match to two different organisms, and yet concretely assigned as one of them. If the query matches multiple references exceeding the annotation threshold, it would make sense for the ASV to not be annotated as either organism, and instead annotated at a broader classification level. Please explain in more detail what was done. This is particularly an issue for short rRNA amplicons – they can be 100% identical across lineages at even the order level (V9 of the 18S rRNA, for example).**

*MEGAN6 interprets blast hits first by applying a minimum threshold ($\geq$80% sequence identity, $\geq$100 bitscore), then takes the top bitscore value for that ASV and only considers blastn hits that are within 2% of that top bitscore value. Within this list of "best" hits, the LCA parameter controls the threshold "percent of hits" to the same organism in order to assign that ASV to that organism. If it was set to 1 (100%) then every top blastn hit would have to be to the same species for that ASV to be assigned to that species. What we found operationally was that there were sometimes mislabeled genbank sequences that caused abundant ASVs to be unassigned. These ASVs often had many hits to reference sequences for the same organism but then also a hit to an organism or environmental sample that was unrelated (and we believe likely misannotated). For*

*these analyses we decided that relaxing that parameter, so that if 80-85% of quality hits were to the same species then that ASV would still be assigned, was the best compromise. In an ideal situation, better curated reference databases would exist (particularly for COI) that would allow us to set this parameter to 100%. Since we are trying to capture such a wide diversity of life across 18S and COI, we felt at the time of analysis that searching genbank NR was still the best available option.*

*Additional text to justify this is added on lines 301-303.*

**How might those salinity additions have impacted cell physiology and community composition? Were there any shifts in the phytoplankton community apparent, for example with the emergence of salt-tolerant species?**

*We did not observe any changes in community composition that could directly be linked to changes in salinity. Given the relatively short duration of the experiment and the fact that the salt brine solution only raised the salinity by about one unit, to a maximum of less than 36 psu, we do not expect that this was a major factor in determining community composition. Furthermore, the samples taken for DNA analysis were integrated water samples taken above the depth of the salt brine additions, so their influence on the community composition should be minimal. The comparison of sampling depths to the depths at which the salt brine solution was injected is noted on lines 182-185.*

**Why was more OMZ water (depleted in NOx) added to mesocosms on Day 11? And why using a different mesh size than the original addition?**

*OMZ water was only added to the mesocosms once, on days 11 and 12. This is detailed in section 2.1, on lines 147-162 (and can also be seen graphically in Fig. 2). Regarding the depleted levels of NOx in the added OMZ water, one of the hypotheses of the mesocosm experiment was that NOx depleted water might stimulate nitrogen fixation.*

**Line 360 – Please include a citation for referring readers to the temperature, chlorophyll and macronutrients in the other mesocosms (apart from M1).**

*A reference to Bach et al. 2020 has been added on line 387.*

**Line 504 – Some discussion on whether amplicon sequence abundance correlates with cell biomass is needed. This is notoriously an issue for organisms with high 18S rRNA copy numbers.**

*We have added the text below, with appropriate references, on lines 535-538. We note the caveats to interpreting amplicon abundance as biomass, but also note that our high read numbers are corroborated by high abundance of Akashiwo as measured using other sampling methods.*

*"While the correlation between cell biomass and amplicon sequence abundance is complicated both by technical biases in metabarcoding and variation in gene copy numbers (Martin et al.,*

*2022), the high biomass of certain taxa as inferred using read numbers is corroborated by other sampling techniques used to sample community composition. In this experiment, A. sanguinea was also identified as the dominant dinoflagellate by imaging flow cytometry and microscopy in mesocosm M1, and in seven of eight mesocosms overall (Bach et al., 2020)."*

**Line 665 – please rephrase, sentence is not clear.**

*We have rewritten the text on lines 700-704 as follows:*

*"While bony fish are the intended target of the MiFish primers, these primers are also capable of detecting other vertebrates due to sequence similarity in the targeted region of 12S rRNA gene, albeit with weaker amplification (Miya et al., 2015; Monuki et al., 2021). As such, while we detected seabirds using our primers, they are likely not well represented in our data; primers that specifically target birds should improve assessments of their eDNA (Ushio et al., 2018)."*

**Comment on bg-2022-212**

Anonymous Referee #2

Referee comment on "Ecological divergence of a mesocosm in an eastern boundary upwelling system assessed with multi-marker environmental DNA metabarcoding" by Markus A. Min et al., Biogeosciences Discuss., https://doi.org/10.5194/bg-2022-212-RC2, 2023

*Thank you for your helpful comments. The responses to the specific concerns you raised are below.*

**Line 85, "(Bach et al., this issue)", in my opinion, writing a reference like this is not suitable. Please refer to the style of the journal and make adjustments.**

*References to this paper have been updated to "Bach et al., 2020."*

**Line 125, make an explanation about the M1-M8 in the figure captions. Line 166-180, this paragraph may be better to put in part "2.7" or "2.8". Line 381, PCA, not RPCA**

*M1-M8 are now addressed in the caption for Figure 1 on line 141.*

*For the paragraph on lines 166-180, we believe that this paragraph fits best in the current section ("2.2 Sample collection"), as it addresses the collection of field control samples. However, we do see the link between this paragraph and 2.7 ("Quality Control and Decontamination"). As such, have updated the sentence on lines 325-326 in section 2.7 to link back to section 2.2*

*The mention of "RPCA" rather than "PCA" on line 381 is intentional, as we are using robust Aitchison PCA (RPCA), which we first introduce and define on line 336.*

**For all the figures, please make them more attractive and uniform in a font size at the x and y-axis.**

*We have replaced all previously bold text with plain text and ensured that all of our axis titles and axis labels are the same size. The ggplot package in R was used to generate all figures; as seen in the code on our GitHub repository, all axis text is size 15, and all axis titles are size 20. Remaining differences in how large the text appears, particularly in the PDF, are due to how large the figures appear when fit into a Word Document. For example, all of the text in Figure 3, which is a large, multi-panel figure, appears much smaller than in Figure 1, which is a simple map. If the differing sizes in text appearance remains an issue when the proofs are compiled for this figure, we are happy to make changes to the font sizes to make them appear as uniform as possible in the final publication.*